# Distinct molecular characteristics and virulence profiles of carbapenem-resistant *Acinetobacter baumannii*, *Escherichia coli*, and *Enterobacter cloacae* isolated from patients with inborn errors of immunity

Xiaodan Zhu,[1] Pan Fu,[2] Songzhen Yang,[1] Wenjie Wang,[1] Wenjing Ying,[1] Bijun Sun,[1] Jinqiao Sun,[1] Chuanqing Wang,[2] Qinhua Zhou,[1] Xiaochuan Wang[1,3]

**ABSTRACT** Bacterial infections, especially multidrug-resistant Gram-negative bacterial infections, pose a great threat to patients with inborn errors of immunity (IEIs). This study investigates the molecular and virulence profiles of carbapenem-resistant *Acinetobacter baumannii* (CR-AB), carbapenem-resistant *Escherichia coli* (CR-ECO), and carbapenem-resistant *Enterobacter cloacae* (CR-ECL) strains from patients with IEI. Strains from IEI and non-IEI groups underwent antimicrobial susceptibility testing and whole-genome sequencing (NovaSeq 6000 PE150), with statistical analysis of differences. A total of 24 CR-AB, 17 CR-ECO, and 16 CR-ECL strains were included. Most CR-AB strains in the IEI group belonged to ST2 (81.8%), all harbored $bla_{OXA-23}$, followed by ST109 ($bla_{OXA-58}$, 9.1%) and ST70 ($bla_{NDM-1}$, 9.1%), whereas all CR-ABs in the non-IEI group were ST2 with $bla_{OXA-23}$. CR-ECL strains from the IEI group harbored $bla_{KPC-2}$ (14.3%) and $bla_{VIM-1}$ (14.3%), in contrast to the non-IEI group. Compared to the non-IEI group, strains in the IEI group exhibited lower carriage of immune modulation genes in CR-AB (18.2%–45.5% vs. 40.0%–80.0%), reduced carriage of adherence genes (50.0%–62.5% vs. 88.9%–100.0%) and nutritional/metabolic factor genes (25.0% vs. 55.6%) in CR-ECO, and lower carriage of nutritional/metabolic factor genes (28.6% vs. 50.0%) in CR-ECL. No statistically significant differences were observed between the groups, except for the *fimA* gene in CR-ECO ($P < 0.05$). The distinct molecular characteristics and reduced virulence gene carriage were observed in CR-AB, CR-ECO, and CR-ECL isolates from patients with IEI, with greater carbapenemase gene diversity in CR-AB and CR-ECL. These findings emphasize the need for enhanced surveillance and tailored antimicrobial strategies in IEI populations.

**IMPORTANCE** Bacterial infection, especially drug-resistant bacterial infection, poses a great risk to patients with IEIs. Antimicrobial resistance, particularly in pediatric patients, is a growing global health threat. Bacteria undergo a series of adaptive changes in response to pressures from the host. Patients with IEI provide a unique immune environment that may profoundly influence the molecular characteristics of bacterial pathogens. However, little is known about the molecular and virulence profiles of carbapenem-resistant *Acinetobacter baumannii* (CR-AB), *Escherichia coli* (CR-ECO), and *Enterobacter cloacae* (CR-ECL) isolated from patients with IEI. This study, the first of its kind, shows that CR-AB, CR-ECO, and CR-ECL from patients with IEI have distinct molecular profiles, including reduced virulence gene carriage and more diverse carbapenemase genes. It highlights the role of host immune status in shaping pathogen evolution and resistance, emphasizing the importance of monitoring the adaptive variation of these bacteria in patients with IEI.

**Peer Reviewers** Innocent Afeke, University of Health and Allied Sciences, Ho, Ghana; Dongsheng Han, The First Affiliated Hospital, Zhejiang University School of Medicine, Hangzhou, Zhejiang, China

Address correspondence to Chuanqing Wang, chuanqing523@163.com, Qinhua Zhou, qinhua_zhou@fudan.edu.cn, or Xiaochuan Wang, xchwang@shmu.edu.cn.

Xiaodan Zhu and Pan Fu contributed equally to this article. The order of co-first authors was determined based on their relative contributions.

The authors declare no conflict of interest.

See the funding table on p. 11.

**KEYWORDS** Inborn errors of immunity, Children, Carbapenem-resistant, *Acinetobacter baumannii*, *Escherichia coli*, *Enterobacter cloacae*

Inborn errors of immunity (IEIs) are a group of disorders caused by single-gene mutations that impair immune function, leading to increased susceptibility to infections, autoimmune diseases, allergies, and malignancies (1). These immune defects hinder pathogen recognition and clearance, making patients with IEI vulnerable to rare, atypical, or opportunistic infections (2, 3). Additionally, the frequent visits to healthcare facilities by patients with IEI increase the risk of cross-hospital infections. Infections, particularly bacterial infections, including Gram-negative bacteria, are one of the major clinical challenges faced by patients with IEI.

Antimicrobial resistance has emerged as a critical global health threat, with multi-drug-resistant organisms (MDROs) causing infections that are increasingly difficult to treat and associated with heightened morbidity and mortality (4). In 2017, the World Health Organization prioritized carbapenem-resistant *Acinetobacter baumannii* (CR-AB), carbapenem-resistant *Escherichia coli* (CR-ECO), and carbapenem-resistant *Enterobacter cloacae* (CR-ECL) as critical pathogens requiring urgent intervention and novel antimicrobial development (5). Treatment options for these resistant pathogens are limited, with even fewer choices available for pediatric patients (6). Alarmingly, studies have shown that MDROs isolated from pediatric patients exhibit widespread resistance, with $bla_{NDM}$ genes prominently disseminated among CR-ECO and CR-ECL strains (7, 8). Immunocompromised patients are closely associated with infections caused by drug-resistant bacteria (9, 10). A previous study reported that bacteria isolated from patients with primary immunodeficiency disorders exhibited higher levels of antimicrobial resistance compared to those from immunocompetent patients (11). Understanding the resistance profiles and underlying mechanisms of CR-AB, CR-ECO, and CR-ECL in children with IEI is therefore essential.

Under antimicrobial pressure, bacteria carrying resistance genes survive, increasing the frequency of these genes within the host population (12). In addition to antimicrobials, bacteria must overcome various challenges within the host, including obtaining nutrients and evading the immune system (13). In experiments involving *Caenorhabditis elegans* nematodes infected with pathogens, it was suggested that immunocompromised hosts exhibit significant genomic diversification and reduced virulence in the pathogens (14). These findings imply that host immune pressure is closely linked to changes in the genetic characteristics of bacteria.

Therefore, we hypothesize that carbapenem-resistant bacteria isolated from patients with IEI possess distinct genetic characteristics and virulence profiles shaped by their impaired immune environment. Currently, there is no related research investigating the molecular features of CR-AB, CR-ECO, and CR-ECL isolated from patients with IEI. This study aims to analyze the molecular characteristics and virulence profiles of strains isolated from patients with IEI between 2017 and 2023, including CR-AB, CR-ECO, and CR-ECL, providing insights into the genetic adaptations of these pathogens and the influence of host immune status on pathogen evolution.

## MATERIALS AND METHODS

### Study design and clinical data

This study included strains isolated from hospitalized patients at Children's Hospital of Fudan University from 2017 to 2023. All strains were identified using matrix-assisted laser desorption/ionization time-of-flight mass spectrometry (Bruker, France), tested for antimicrobial susceptibility, and stored at −80°C. The strains were categorized into non-IEI and IEI groups based on the source patient. Non-IEI group: strains were randomly selected from patients without IEI. To reduce the potential influence of hospital-acquired infections on the analyzed strains, those isolated from pediatric intensive care unit (PICU) settings were excluded from the study. In the IEI group, strains were obtained

from patients with IEI, also excluding PICU sources. IEI was diagnosed based on clinical manifestations, immunological tests, and genetic testing, with classification following the IUIS 2022 guidelines (1).

Clinical data for all patients were collected, including age at admission, gender, inpatient days, IEI classifications, site of bacterial isolation, and outcomes. The immunological phenotypes and genetic results for patients with IEI were also recorded. The immunological phenotype was assessed during the patient's hospitalization at the time of strain isolation.

For antimicrobial susceptibility and gene-carrying rate analysis, strains isolated from the same site of the same patient were selected based on the first isolation for analysis. In the phylogenetic tree analysis, all strains were included in the analysis. CR-AB isolates were defined as resistant to imipenem or meropenem, while CR-ECO and CR-ECL isolates were defined as resistant to at least one of the carbapenem agents, including ertapenem, imipenem, or meropenem.

Due to the rarity of IEI diseases, there was heterogeneity among the types of patients with IEI included in this study. However, we also summarized the virulence and resistance genes of strains isolated from the predominant IEI types. Specifically, CR-AB was isolated from patients with combined immunodeficiency (CID), while CR-ECO and CR-ECL were isolated from patients with chronic granulomatous disease (CGD).

## Antimicrobial susceptibility test

All strains were tested for antimicrobial susceptibility using the VITEK 2 Compact automated microbiology system with AST-GN13 cards (bioMérieux). For antimicrobials not covered by the AST-GN13 card, the Kirby-Bauer test was used to obtain susceptibility results. All results were interpreted according to the 2024 breakpoints established by the Clinical and Laboratory Standards Institute (M100-S34) (15). The following reference strains were used to ensure the reproducibility of the antibiotic susceptibility testing procedure: *Staphylococcus aureus* American Type Culture Collection (ATCC) 25923, *Enterococcus faecalis* ATCC 29212, *Escherichia coli* ATCC 25922, and *Pseudomonas aeruginosa* ATCC 27853.

## DNA extraction and whole-genome sequencing

Frozen strains were streaked onto blood agar plates and incubated overnight at 35°C for 16–18 hours. Before DNA extraction, the strains were subcultured twice. DNA extraction was performed according to the kit instructions (Yeasen, Shanghai, China). Subsequently, the concentration and quality of DNA were assessed. Qualified DNA samples were used for library construction, purification, and recovery. After quantification, the libraries were sequenced using the NovaSeq 6000 platform with PE150 reads.

## Assembly and annotation

Reads were demultiplexed based on barcode information, and quality filtering was performed on the raw reads obtained from sequencing using fastp (version 0.23.1) to generate high-quality clean reads. The clean reads were assembled using SPAdes (version 3.13.0) with default parameters, and the assembly quality was assessed using QUAST (version 5.0.2). Gene prediction for the scaffolds was performed using Prokka (version 1.14.6), generating standardized annotation results. Genome assembly quality metrics for each strain are provided in Table S1.

## Multilocus sequence typing and core-pan analysis

Multilocus sequence typing (MLST) was performed using MLST (https://github.com/tseemann/mlst) and the PubMLST database. A*cinetobacter baumannii* ST typing was conducted using the Pasteur scheme, while *Escherichia coli* ST typing followed the Achtman scheme. Core-pan genes were extracted using Roary to calculate the GFF files from Prokka results. Antimicrobial resistance genes were identified through comparison

with the Comprehensive Antibiotic Resistance Database (https://card.mcmaster.ca/), and virulence genes were identified by comparison with the Virulence Factor Database (http://www.mgc.ac.cn/cgi-bin/VFs/v5/main.cgi). Assembled sequences were aligned to the reference genome using NUCmer (version 3.1) to extract core gene regions and identify single-nucleotide polymorphism (SNP) sites. Then, evolutionary distances were calculated using the generalized time-reversible model, and a maximum likelihood tree was constructed using a combination of heuristic methods. Bootstrap analysis was conducted using 1,000 replicates. The phylogenetic tree was visualized with Tree Visulization By One Table (https://www.chiplot.online/tvbot.html).

## Statistical analysis

Statistical analysis of data was performed using SPSS software (version 27.0). Continuous variables were described using the median and interquartile range. Categorical variables were expressed as frequencies (percentages). A *t*-test or Fisher's exact test was used to compare differences between the two groups. A *P* value less than 0.05 represented a statistically significant difference.

## RESULTS

### Demographic characteristics of patients harboring CR-AB, CR-ECO, and CR-ECL

The CR-AB, CR-ECO, and CR-ECL strains were mainly isolated from male patients in both groups. Among the three strains, patients with IEI had longer hospital stays compared to those in the non-IEI group (47 vs. 34, 39 vs. 11, and 37 vs. 27), with a statistically significant difference observed in CR-ECO ($P < 0.05$). The main classification of IEI for CR-AB was combined immunodeficiencies (55.6%), while for both CR-ECO and CR-ECL, it was congenital defects of phagocytes (75.0% and 71.4%). For CR-AB, the respiratory tract was the primary source of bacterial isolates in both groups, with 50.0% of isolates from the non-IEI group and 78.6% from the IEI group. In contrast, for CR-ECO and CR-ECL, the main sources in the IEI group were the respiratory tract (25.0% and 50.0%) and skin and soft tissue infections (37.5% and 25.0%), while the non-IEI group primarily had isolates from the urinary tract (77.8% and 75.0%). The use of antifungal and antiviral medications was significantly higher in the IEI group for CR-AB ($P < 0.01$ and $P < 0.001$) and CR-ECO ($P < 0.01$ and $P < 0.05$) strains. The use of antifungal medications was higher in the IEI group for CR-ECL ($P < 0.001$), as shown in Table 1. Detailed clinical data of the patients are shown in Tables S2 and S3.

### Antimicrobial resistance of CR-AB, CR-ECO, and CR-ECL isolates

The antimicrobial resistance rates of CR-AB isolates to gentamicin (72.7% vs. 100.0%) and levofloxacin (36.4% vs. 60.0%) were lower in the IEI group compared to the non-IEI group. The resistance rates of CR-ECO isolates to levofloxacin (75.0% vs. 44.4%) and trimethoprim-sulfamethoxazole (TMP-SMX) (100.0% vs. 55.6%) were higher in the IEI group compared to the non-IEI group. In contrast, the resistance rates of CR-ECL isolates to gentamicin (14.3% vs. 50.0%) and levofloxacin (14.3% vs. 37.5%) were lower in the IEI group than in the non-IEI group, while resistance to TMP-SMX was higher in the IEI group (85.7% vs. 37.5%) (in Fig. 1). The above analysis was descriptive and had no statistical significance.

### MLST typing and phylogenetic analysis of CR-AB, CR-ECO, and CR-ECL strains

For CR-AB, ST2 was the predominant ST type, with 81.8% of strains from the IEI group and 100.0% from the non-IEI group. The IEI group had only two strains with non-ST2 types, namely, ST70 and ST109. The ST types for CR-ECO were dispersed, with the IEI group predominantly having ST167 (*n* = 2, 28.6%) and ST410 (*n* = 2, 28.6%), while the non-IEI group mainly had ST38 (*n* = 2, 22.2%) and ST410 (*n* = 2, 22.2%). CR-ECL showed a

TABLE 1 Demographic and clinical characteristics of CR-AB, CR-ECO, and CR-ECL isolated from patients of IEI and non-IEI group[a]

| Characteristics | CR-AB | | P value | CR-ECO | | P value | CR-ECL | | P value |
|---|---|---|---|---|---|---|---|---|---|
| | Non-IEI (n = 10) | IEI (n = 9) | | Non-IEI (n = 9) | IEI (n = 8) | | Non-IEI (n = 8) | IEI (n = 7) | |
| Age (months), median (IQR) | 25.0 (19.3–37.3) | 9.2 (4.1–24.0) | 0.623 | 7.2 (5.2–16.0) | 18.5 (8.6–31.0) | 0.465 | 8.6 (1.2–58.0) | 19.0 (9.9–36.5) | 0.669 |
| Gender (M/F) | 7/3 | 8/1 | 0.582 | 7/2 | 7/1 | 1.00 | 5/3 | 7/0 | 0.200 |
| IEI classifications | | | | | | | | | |
| Combined immunodeficiencies | – | 5 | – | – | 1 | – | – | 2 | – |
| Predominantly antibody deficiencies | – | 0 | – | – | 1 | – | – | 0 | – |
| Congenital defects of phagocytes | – | 3 | – | – | 6 | – | – | 5 | – |
| Defects in intrinsic and innate immunity | – | 1 | – | – | 0 | – | – | 0 | – |
| Inpatient days, median | 34 | 47 | 0.988 | 11 | 39 | 0.013 | 27 | 37 | 0.755 |
| Outcome at discharge | | | | | | | | | |
| Recovery | 9 | 6 | – | 9 | 7 | – | 8 | 5 | – |
| Death | 1 | 3 | – | 0 | 1 | – | 0 | 2 | – |
| Number of isolated strains | 10 | 14 | – | 9 | 8 | – | 8 | 8 | – |
| Isolation sites, n (%) | | | | | | | | | |
| Respiratory tract | 5 (50.0) | 11 (78.6) | 0.204 | 2 (22.2) | 2 (25.0) | 1.000 | 1 (12.5) | 4 (50.0) | 0.282 |
| SSTIs | 2 (20.0) | 2 (14.3) | 1.000 | 0 (0.0) | 3 (37.5) | 0.082 | 1 (12.5) | 2 (25.0) | 1.000 |
| Bloodstream | 2 (20.0) | 1 (7.1) | 0.550 | 0 (0.0) | 1 (12.5) | 0.471 | 0 (0.0) | 0 (0.0) | – |
| Bile | 1 (10.0) | 0 (0.0) | 0.417 | 0 (0.0) | 0 (0.0) | – | 0 (0.0) | 0 (0.0) | – |
| Pleural drainage | 0 (0.0) | 0 (0.0) | – | 0 (0.0) | 0 (0.0) | – | 0 (0.0) | 1 (12.5) | 1.000 |
| Urinary tract | 0 (0.0) | 0 (0.0) | – | 7 (77.8) | 2 (25.0) | 0.057 | 6 (75.0) | 1 (12.5) | 0.041 |
| Anti-infective drugs, n (%) | | | | | | | | | |
| Combined with antifungal drugs | 3 (30.0) | 9 (100.0) | 0.003 | 1 (11.1) | 7 (87.5) | 0.003 | 0 (0.0) | 7 (100.0) | <0.001 |
| Combined with antiviral drugs | 0 (0.0) | 7 (77.8) | <0.001 | 0 (0.0) | 4 (50.0) | 0.029 | 1 (12.5) | 4 (57.1) | 0.120 |

[a]Abbreviations: IQR, interquartile range; M/F, male/female; SSTI, skin and soft tissue infection; "-" stands for "Not applicable".

scattered distribution of ST types, with the non-IEI group primarily consisting of ST127 (n = 2, 25.0%). Phylogenetic tree analyses for all three bacterial species indicated that the genetic differences between the two groups were not significant (see Fig. 2).

## Antibiotic resistance and virulence gene profiles of CR-AB strains

Given the small sample size, comparisons were primarily descriptive, with statistically significant differences indicated. The comparison of isolated strains was conducted not only between IEI and non-IEI groups, but also among subgroups of patients with IEI, categorized based on the major IEI types. The detection of resistance genes revealed that most CR-AB strains in the IEI group belonged to ST2 (81.8%); all harbored $bla_{OXA-23}$, followed by ST109 ($bla_{OXA-58}$, 9.1%) and ST70 ($bla_{NDM-1}$, 9.1%), whereas all CR-ABs in the non-IEI group were ST2 with $bla_{OXA-23}$. All strains tested positive for the $bla_{ADC}$ gene. Furthermore, the carriage rate of $bla_{TEM-1}$ was lower in the IEI group compared to the non-IEI group (54.6% vs. 90.0%). Regarding aminoglycoside resistance genes, the carriage rates of AAC (3)-Ia, APH (3′)-Ib, and APH (6)-Id were lower in the IEI group (9.1%–54.6% vs. 40.0%–70.0%). For quinolone resistance genes, the AdeFGH efflux system genes adeF, adeG, and adeH were detected in 100% of strains from both groups. In terms of virulence genes, strains from both the IEI and non-IEI groups carried a variety of virulence factors associated with adhesion, biofilm formation, exotoxins, immune modulation, and nutritional/metabolic factors. The carriage rates of the immune

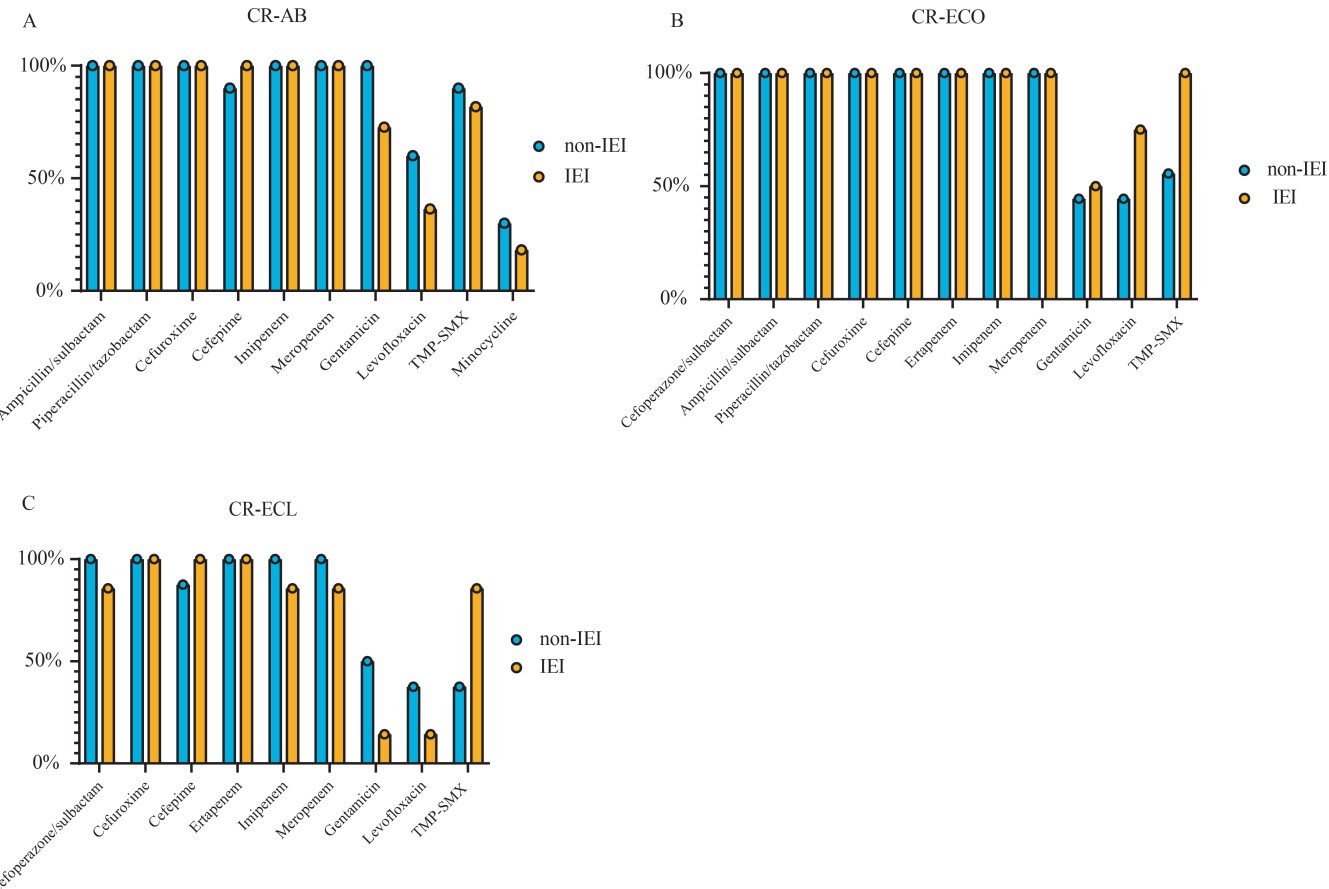

**FIG 1** Comparison of antimicrobial resistance rates for CR-AB (A), CR-ECO (B), and CR-ECL (C). TMP-SMX, trimethoprim-sulfamethoxazole.

regulation genes *ACICU_RS00475* (18.2% vs. 40.0%) and *ACICU_RS00485* (45.5% vs. 80.0%) were lower in the IEI group compared to the non-IEI group (in Fig. 3A; Fig. S1A).

## Antibiotic resistance and virulence gene profiles of CR-ECO strains

Detection of resistance genes revealed that both groups of CR-ECO strains exhibited a 100% carriage rate of the $bla_{NDM-1}$ gene. In the IEI group, the predominant $bla_{CTX-M}$ gene was $bla_{CTX-M-15}$ (62.5%), whereas $bla_{CTX-M-14}$ was more common in the non-IEI group (44.4%). Sulfonamide resistance genes, particularly *sul1* (100.0% vs. 55.6%) and *sul2* (75.0% vs. 44.4%), were more prevalent in the IEI group. Virulence gene detection revealed that the carriage rates of type 1 fimbriae-related genes were lower in the IEI group, with *fimA* (50.0% vs. 100.0%, $P < 0.05$), *fimC* (50.0% vs. 88.9%), and *fimH* (62.5% vs. 88.9%). Virulence genes associated with invasion *kpsM*, *kpsS*, *kpsC*, and *kpsD* were lower (12.5%–25.0% vs. 55.6%) in the IEI group compared to the non-IEI group. The carriage rates of the effector delivery system genes, including *gspG*, *gspH*, *gspI*, and *gspJ*, were lower (20.0% vs. 66.7%) in CGD patients. Virulence genes associated with nutritional/ metabolic factors, including *chuA*, *chuT*, *chuU*, and *chuV*, were also less common (25.0% vs. 55.6%) in strains from the IEI group, and a similar trend could be seen in the isolates of CGD patients (see Fig. 3B; Fig. S1B).

## Antibiotic resistance and virulence gene profiles of CR-ECL strains

Resistance gene detection in CR-ECL strains revealed that the majority of strains in both groups carried the $bla_{NDM-1}$ gene. In the IEI group, the prevalence of the β-lactamase gene included $bla_{NDM-1}$ (42.9%), $bla_{KPC-2}$ (14.3%), $bla_{VIM-1}$ (14.3%), $bla_{IMP-26}$ (14.3%), and $bla_{CTX-M-15}$ (71.4%). In the non-IEI group, the prevalence of β-lactamase genes included

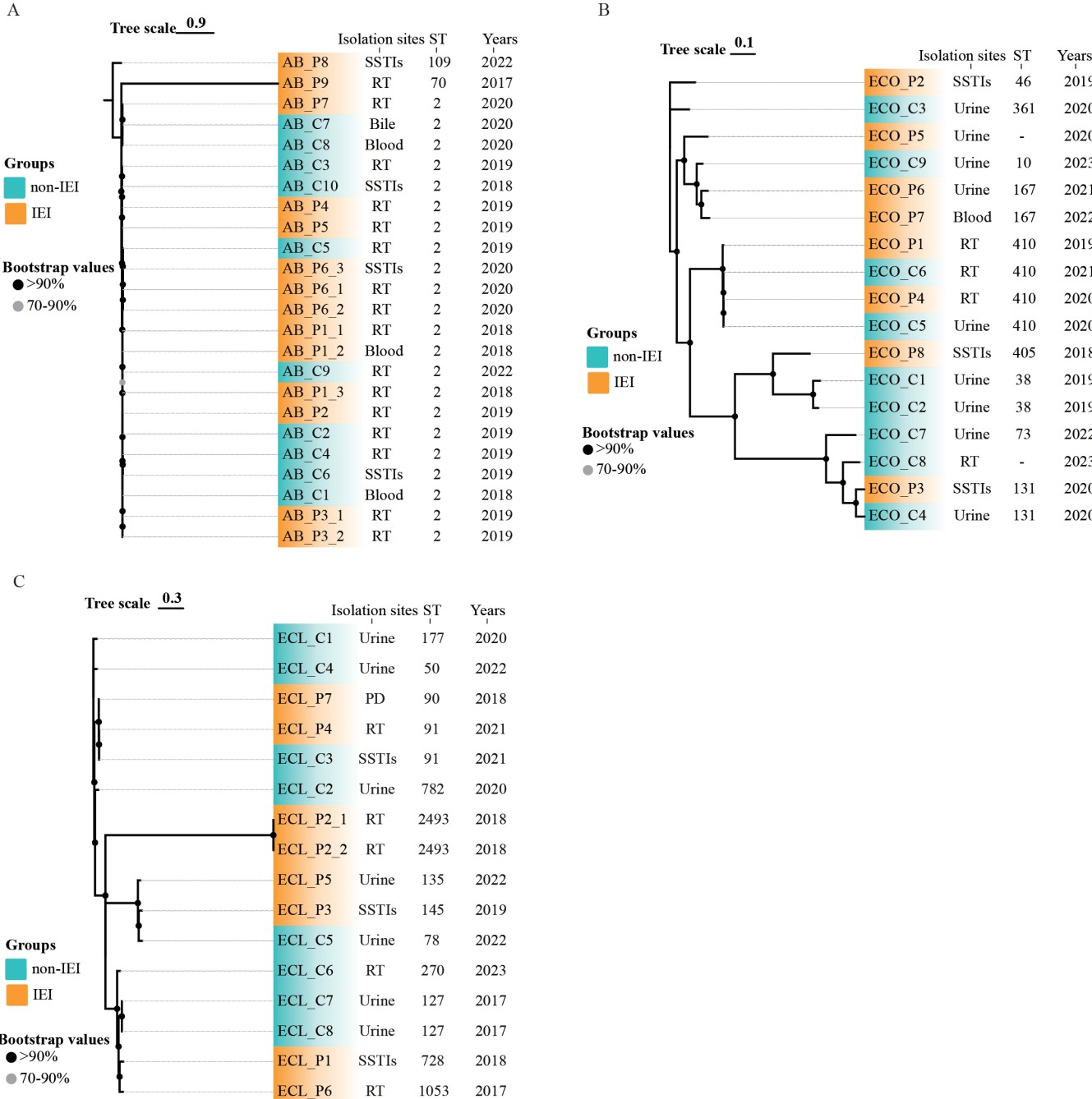

**FIG 2** Phylogenetic tree of bacterial strain genomes constructed from core single-nucleotide polymorphisms. (A) Phylogenetic tree of CR-AB strains, (B) phylogenetic tree of CR-ECO strains, and (C) phylogenetic tree of CR-ECL strains. PD, pleural drainage; RT, respiratory tract; SSTI, skin and soft tissue infection; "-" indicates unknown ST type.

$bla_{NDM-1}$ (62.5%), $bla_{IMP-26}$ (37.5%), and $bla_{CTX-M-15}$ (25.0%). The carriage rates of aminoglycoside resistance genes were higher in the IEI group (28.6%–85.7% vs. 0.0%–25.0%), including *AAC (6′)-Ib-cr6*, *aadA2*, *ANT (2′′)-Ia* and *APH (3′)-Ib*. Among these, *AAC (6′)-Ib-cr6* showed a statistically significant difference ($P < 0.05$). Additionally, the macrolide resistance gene *mphA* was more prevalent in the IEI group (85.7% vs. 25.0%, $P < 0.05$). In the category of nutritional/metabolic factors, the genes *iroC*, *iroB*, and *iroN* showed a slightly lower (28.6% vs. 50.0%) prevalence in the IEI group, and a similar trend could be seen in the isolates of CGD patients (see Fig. 3C; Fig. S1C).

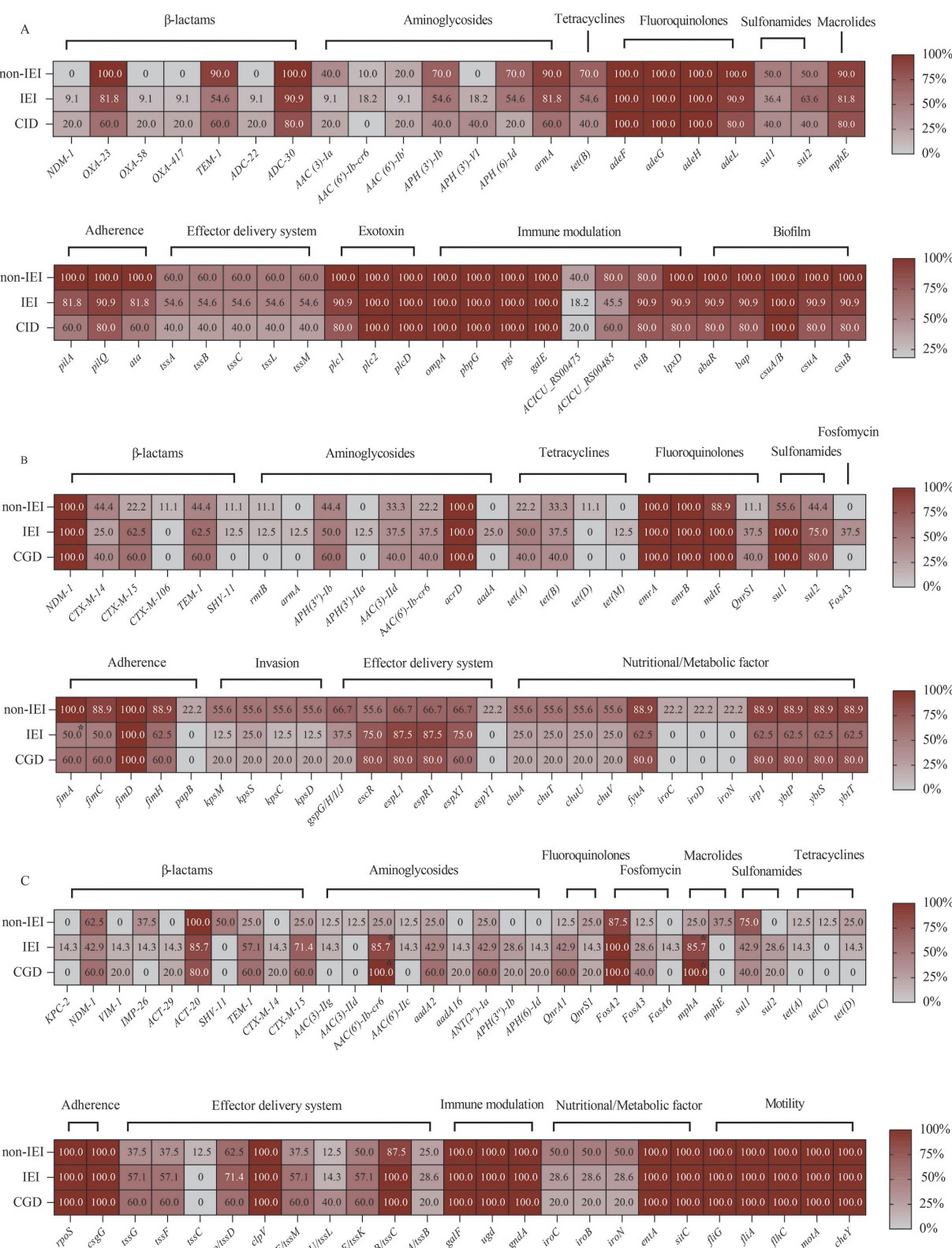

**FIG 3** A heatmap comparison of antibiotic resistance and virulence gene detection rates between the IEI and non-IEI groups. (A) Antibiotic resistance and virulence genes in CR-AB, (B) antibiotic resistance and virulence genes in CR-ECO, and (C) antibiotic resistance and virulence genes in CR-ECL. Darker colors indicate higher detection rates. *, *P* < 0.05.

## DISCUSSION

MDRO infections have rapidly emerged as a critical global public health threat, particularly among vulnerable populations such as children (8, 16, 17). The combination of high transmissibility and extensive antimicrobial resistance in these pathogens poses severe challenges to infection management and control. For patients with IEI, the inherent immune defects make them particularly vulnerable to highly resistant bacterial infections, which not only complicate treatment but also lead to more severe clinical outcomes. This study represents the first investigation into the genetic characteristics of CR-AB, CR-ECO, and CR-ECL isolates from patients with IEI. CR-AB isolates from children mainly carried the $bla_{OXA-23}$ gene, while CR-ECO and CR-ECL primarily harbored the $bla_{NDM-1}$ gene. However, carbapenemase genes were more diverse in CR-AB and CR-ECL isolates from patients with IEI. Additionally, the prevalence of virulence genes in isolates from patients with IEI was slightly lower.

Given the differences in antimicrobial use and immune pressure between patients of IEI and non-IEI group, we hypothesize that there may be genetic background differences between the bacterial strains isolated from the two groups. In the core-SNP phylogenetic tree analysis, the strains isolated from the IEI group and the non-IEI group did not show significant separation, which may be attributed to the high genetic similarity between the two groups and the small sample size (18). This also implies that these resistant strains may have horizontal gene transfer between different populations. Additionally, since core-SNP analysis focuses on the core genome, some minor mutations and variations may not be captured.

*Acinetobacter baumannii* is a significant opportunistic pathogen in nosocomial infections, particularly affecting patients with compromised immune systems, and is associated with high rates of complications and mortality (19–21). In this study, CR-AB was primarily detected in patients with CID, which is related to the increased susceptibility to severe infections due to reduced T-cell immunity in these individuals (22). Conversely, CR-ECO and CR-ECL were predominantly detected in patients with CGD, a condition associated with heightened vulnerability to catalase-positive organisms (23). Notably, patients with IEI in this study had significantly longer hospital stays than patients without IEI. Patients with IEI often experience specific infections, including viral, fungal, and mycobacterial infections. Additionally, some patients may present with other severe conditions, such as severe pneumonia, acute respiratory distress syndrome, respiratory failure, and sepsis. As a result, bacterial infections may not have been the primary cause of prolonged hospitalization and were therefore not further explored in this study.

Antimicrobial resistance in the CR-ECO and CR-ECL isolates from patients with IEI was particularly notable for TMP-SMX, which may be explained by the frequent use of this drug in CGD patients for the treatment and prevention of *Pneumocystis* pneumonia (24). In the analysis of TMP-SMX resistance-related genes, we utilized both protein-based and gene-based alignment methods. It was observed that CR-ECO isolates from patients with IEI were uniformly resistant and exhibited a significantly higher prevalence of the *sul1* gene compared to isolates from patients without IEI. Furthermore, all TMP-SMX-resistant CR-ECL isolates from patients with IEI were found to carry the *sul1* gene. These findings indicate that the isolates from patients with IEI may have undergone chronic infection and adaptive changes, potentially contributing to their increased resistance to sulfonamide antibiotics.

Carbapenem resistance, the focus of this study, is primarily driven by carbapenemase production (25, 26). These enzymes are classified into three major groups: (i) class A serine β-lactamases (e.g., $bla_{KPC}$); (ii) class B metallo-β-lactamases (e.g., $bla_{NDM}$, $bla_{VIM}$, and $bla_{IMP}$); and (iii) class D serine β-lactamases (e.g., $bla_{OXA}$) (27). Additional mechanisms, including porin loss and efflux pump overexpression, also contribute to the resistance phenotype (28). In this study, ST2 was the predominant sequence type among CR-AB isolates, and $bla_{OXA-23}$ was the most common carbapenemase gene, consistent with global trends (20). Interestingly, two CR-AB isolates from patients with IEI harbored

$bla_{NDM-1}$ and $bla_{OXA-58}$, associated with ST70 and ST109, respectively. For CR-ECO, all isolates carried $bla_{NDM-1}$, aligning with prior reports of its global prevalence in CR-ECO (8, 29, 30). CR-ECL isolates from both groups predominantly carried $bla_{NDM-1}$, though IEI-associated isolates exhibited greater diversity in carbapenemase genes, including $bla_{KPC-2}$ and $bla_{VIM-1}$. In addition, we found that the resistance genes *AAC (6′)-lb-cr6* and *mphA* carried by CR-ECL isolates from the IEI group were significantly higher than those in the non-IEI group. This diversity likely reflects the frequent and prolonged hospitalizations of patients with IEI, which increase the risk of acquiring resistant genes through cross-transmission (31–33).

In the interaction between pathogens and hosts, limited resources often interfere with the maintenance of pathogen virulence (34, 35). Moreover, antimicrobials not only drive changes in bacterial resistance but also influence the evolution of bacterial virulence (34). Experimental models further highlight the interplay between host immunity and pathogen virulence. In one study, *Plasmodium chabaudi* was serially passaged in CD4$^+$ T cell-depleted and control mice, revealing that parasites from the CD4$^+$ T cell-depleted group evolved greater virulence compared to those from the control group (36). Similarly, in a malaria model, parasites passaged in immunized (previously infected) mice exhibited enhanced virulence compared to those evolved in non-immunized mice (37). These findings collectively underscore that host immunity can exert selective pressures on pathogens, influencing their virulence evolution and adaptive strategies. In this study, the immune regulation genes with reduced carriage in CR-AB isolates from patients with IEI were primarily capsule formation-related genes, which are closely associated with bacterial resistance to phagocytosis. The carriage rates of adherence and nutritional/metabolic factor genes in CR-ECO, as well as nutrition/metabolism-related genes in CR-ECL, were all lower than those in the non-IEI group. The phenomenon of reduced carriage of virulence genes was also observed in the IEI subgroup. A similar phenomenon has also been observed in studies of chronic respiratory infections in cystic fibrosis patients. *Pseudomonas aeruginosa* often undergoes gene loss, particularly in virulence-related genes, to evade the host immune response (38). The reduced carriage of virulence genes in isolates from patients with IEI may reflect an adaptive strategy of these pathogens to persist in immunocompromised hosts. These changes are likely driven by the complex selective pressures in patients with IEI, including altered immune pressure and the impact of antibiotic exposure.

This study has several limitations. First, the rarity of IEI and the inclusion of carbapenem-resistant strains resulted in a small and heterogeneous sample size. However, this is the first detailed analysis of carbapenem-resistant strains in the IEI population, providing valuable data on the prevalence of resistant strains within this group and offering preliminary theoretical insights into the clinical management of carbapenem-resistant infections in patients with IEI. Additionally, although the control group also contained carbapenem-resistant strains, these isolates were not obtained from PICU patients, which limited the ability to strictly match variables such as age, gender, and isolation site between the IEI and control groups. Finally, the retrospective design of the study precluded a precise assessment of the pathogenicity of certain strains.

In conclusion, the CR-AB and CR-ECO isolates from patients with IEI demonstrated a greater diversity of carbapenemase genes, along with notably higher sulfonamide resistance in CR-ECO and CR-ECL strains. Nonetheless, the carriage rates of virulence genes were lower among bacteria isolated from patients with IEI. This study emphasizes the critical role of host immune status in shaping pathogen evolution and resistance mechanisms. Future research should investigate how these genetic adaptations influence the persistence and pathogenicity of bacteria in immunodeficient hosts, paving the way for tailored therapeutic strategies for patients with IEI.

## ACKNOWLEDGMENTS

We sincerely thank the medical staff of the Clinical Microbiology Laboratory for their invaluable assistance with bacterial strain testing and preservation.

This study was supported by grants from the Project supported by Shanghai Municipal Science and Technology Major Project (ZD2021CY001) and the National Natural Science Foundation of China (81501419).

X.C.W., Q.H.Z., and C.Q.W. conceived and designed the study. X.D.Z. and S.Z.Y. collected data from electronic medical charts and conducted bacterial experiments. X.D.Z. and P.F. analyzed the data. X.C.W., J.Q.S., Q.H.Z., W.J.W., W.J.Y., and B.J.S. diagnosed, treated, and followed up these patients with inborn errors of immunity. P.F. and colleagues from the Department of Clinical Microbiology Laboratory performed the bacteria identification and antimicrobial resistance test, and C.Q.W. supervised the whole process of bacterial experiments. X.D.Z., P.F., and Q.H.Z. wrote the first draft. All the authors discussed, revised, and approved the manuscript.

## AUTHOR AFFILIATIONS

[1]Department of Clinical Immunology, Children's Hospital of Fudan University, National Children's Medical Center, Shanghai, China
[2]Department of Clinical Microbiology Laboratory, Children's Hospital of Fudan University, National Children's Medical Center, Shanghai, China
[3]Shanghai Institute of Infectious Disease and Biosecurity, Shanghai, China

## AUTHOR ORCIDs

Pan Fu http://orcid.org/0000-0003-2587-4348
Chuanqing Wang http://orcid.org/0000-0002-0158-9094
Qinhua Zhou http://orcid.org/0000-0002-5112-3727
Xiaochuan Wang http://orcid.org/0000-0003-3768-324X

## FUNDING

| Funder | Grant(s) | Author(s) |
| --- | --- | --- |
| Shanghai Science and Technology Development Foundation | ZD2021CY001 | Xiaochuan Wang |
| National Natural Science Foundation of China | 81501419 | Qinhua Zhou |

## AUTHOR CONTRIBUTIONS

Xiaodan Zhu, Data curation, Formal analysis, Investigation, Methodology, Visualization, Writing – original draft, Writing – review and editing | Pan Fu, Investigation, Methodology, Writing – original draft, Writing – review and editing, Data curation, Formal analysis, Visualization | Songzhen Yang, Investigation, Writing – review and editing | Wenjie Wang, Methodology, Writing – review and editing | Wenjing Ying, Methodology, Writing – review and editing | Bijun Sun, Methodology, Writing – review and editing | Jinqiao Sun, Methodology, Writing – review and editing | Chuanqing Wang, Conceptualization, Supervision, Writing – review and editing | Qinhua Zhou, Conceptualization, Investigation, Methodology, Supervision, Writing – original draft, Writing – review and editing | Xiaochuan Wang, Conceptualization, Funding acquisition, Investigation, Supervision, Writing – review and editing

## ETHICS APPROVAL

The study was approved by the ethics Committee of the Children's Hospital of Fudan University ([No.(2022) 100]).

## ADDITIONAL FILES

The following material is available online.

## Supplemental Material

**Supplemental material (Spectrum00281-25-s0001.docx).** Figure S1; Tables S1 to S3.

## Open Peer Review

**PEER REVIEW HISTORY (review-history.pdf).** An accounting of the reviewer comments and feedback.

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
