## [Reviewer comments · Microbiology Spectrum]

Microbiology Spectrum

Distinct Molecular Characteristics and Virulence Profiles of Carbapenem-Resistant *Acinetobacter baumannii*, *Escherichia coli*, and *Enterobacter cloacae* isolated from Patients with Inborn Errors of Immunity

Xiaodan Zhu, Pan Fu, Songzhen Yang, Wenjie Wang, Wenjing Ying, Bijun Sun, Jinqiao Sun, Chuanqing Wang, Qinhua Zhou, and Xiaochuan Wang

Corresponding Author(s): Qinhua Zhou, Children's Hospital of Fudan University

Review Timeline:

Submission Date:	February 2, 2025
Editorial Decision:	March 25, 2025
Revision Received:	April 10, 2025
Accepted:	May 13, 2025

Editor: Bobby Warren

Reviewer(s): Disclosure of reviewer identity is with reference to reviewer comments included in decision letter(s). The following individuals involved in review of your submission have agreed to reveal their identity: Innocent Afeke (Reviewer #1); Dongsheng Han (Reviewer #3)

Transaction Report:

DOI: <https://doi.org/10.1128/spectrum.00281-25>

Re: Spectrum00281-25 (Distinct Molecular Characteristics and Virulence Profiles of Carbapenem-Resistant *Acinetobacter baumannii*, *Escherichia coli*, and *Enterobacter cloacae* isolated from Patients with Inborn Errors of Immunity)

Dear Dr. Qinhua Zhou:

Thank you for the privilege of reviewing your work. Below you will find my comments, instructions from the Spectrum editorial office, and the reviewer comments.

Revision Guidelines

Sincerely,
Bobby Warren
Editor
Microbiology Spectrum

Reviewer #1 (Comments for the Author):

This study offers a valuable contribution by uncovering how carbapenem-resistant pathogens adapt to and persist in patients with inborn errors of immunity. The integration of host immunological data with microbial genomic analysis is a strong innovation. The manuscript is well written. However, I have the following concerns.

Abstract

- Why is it important to study these bacteria in IEI patients? I suggest that a brief mention of clinical relevance would strengthen the rationale.
- The authors should give specific WGS techniques or platforms used.
- The result section would benefit from mentioning any statistical analyses used to support the comparisons. Were the observed differences statistically significant?

Introduction

- I suggest that the authors introduce IEI earlier and clarify how pediatric patients with IEI fit into the broader context of AMR.
- The hypothesis can be made more specific than just general bacteria. I suggest a more specific one like 'we hypothesize that carbapenem-resistant bacteria isolated from IEI patients possess distinct genetic characteristics and virulence profiles shaped by their impaired immune environment'.

Methods

- The authors acknowledge heterogeneity among IEI patients, but they do not mention any stratification or adjustment methods to account for this variability in statistical analyses.
- The antimicrobial susceptibility testing section lacks details on internal quality controls. Were standard quality control strains (e.g., *E. coli* ATCC 25922, *P. aeruginosa* ATCC 27853) included to verify susceptibility testing accuracy?
- I suggest that the authors should report on assembly quality metrics (e.g., N50 values, genome coverage, contig counts) since these would provide insight into data completeness and accuracy.
- The robustness of phylogenetic trees is unclear-were bootstrap analyses performed to support tree topology?
- Was a power calculation performed to determine the sample size needed to detect significant differences between IEI and non-IEI groups?

Results

- Some p-values are presented vaguely (e.g., $p < 0.01$), while others are precise ($p = 0.013$). Use a consistent format, preferably providing exact p-values when possible.

Reviewer #3 (Comments for the Author):

Zhu X et al. report on the molecular characteristics and virulence profiles of carbapenem-resistant *Acinetobacter baumannii*, *Escherichia coli*, and *Enterobacter cloacae* isolated from patients with inborn errors of immunity (IEI). Their findings indicate that strains from IEI patients exhibit distinct molecular features, including reduced virulence gene carriage and a more diverse repertoire of carbapenemase genes. While the study provides valuable insights, some aspects require further refinement.

1. The biological significance of the reduced virulence gene carriage in IEI strains remains speculative-could this be an adaptation to immunocompromised hosts or a trade-off with antimicrobial resistance?
2. The term "immune modulation genes" in CR-AB is somewhat vague. Clarifying the specific pathways or functions these genes are involved in would enhance understanding.
3. The authors acknowledge the heterogeneity of IEI patients included in this study due to the rarity of these diseases. However, it is unclear why CR-AB isolates were specifically analyzed from CID patients, while CR-ECO and CR-ECL were examined in CGD patients. Was this selection based on the prevalence of these bacterial species in each patient group, or were there specific clinical or microbiological reasons for this categorization?
4. It is worth noting that the length of hospital stay for IEI patients was significantly longer than that of non-IEI patients. However, considering the complex underlying conditions in the IEI group, bacterial infections may not have been the primary reason for the extended hospital stay. I would suggest that the authors briefly explain what the possible reasons might be for the longer hospital stay in the IEI group. This would help clarify whether the extended duration was primarily due to the severity of the underlying immune deficiency or other factors.
5. The way percentage ranges are presented in the manuscript (e.g., ((18.2~45.5)% vs. (40.0~80.0)%)) is unconventional and could lead to confusion. To improve clarity, I recommend using a more standard notation, such as (18.2-45.5% vs. 40.0-80.0%).
6. I noticed an inconsistency in the formatting of p-values between the main text and Table 1. In the text, the p-values are presented in lowercase and non-italicized, while in Table 1, they are capitalized and italicized. Please unify the format.

Dear editors and reviewers

On behalf of my co-authors, I would like to sincerely thank you for the opportunity to revise our manuscript. We greatly appreciate you and the reviewers for your positive and constructive comments and suggestions on our manuscript, titled "Distinct Molecular Characteristics and Virulence Profiles of Carbapenem-Resistant *Acinetobacter baumannii*, *Escherichia coli*, and *Enterobacter cloacae* Isolated from Patients with Inborn Errors of Immunity" (Spectrum00281-25), which we hope will be considered for publication in *Microbiology Spectrum*.

We sincerely thank the reviewers for their thorough review of our manuscript and for providing constructive comments. We greatly appreciate these insightful suggestions, which have been invaluable in guiding the revision and improvement of our work. We have made every effort to address the comments point by point and have revised the manuscript accordingly. A "Marked-Up Manuscript" has been prepared using Word's Track Changes mode, highlighting all modifications made to the original version. Additionally, a "Clean Version" of the manuscript, with all changes accepted, has also been provided for final review. Our responses to the reviewers' comments are presented below. We sincerely hope that our revisions and responses meet your and the reviewers' expectations.

Thank you and best regards.

Yours sincerely,

Qinhua Zhou (in the name of all co-corresponding authors)

Associate chief physician

Department of Clinical Immunology

Children's Hospital of Fudan University

Address: 399 Wanyuan Road, Shanghai, China.

Postcode: 201102

Telephone number: 86-21-64931085

Facsimile number: 86-21-64931901

E-mail: qinhua_zhou@fudan.edu.cn

Author's response to editor and reviewers

Responds to the Reviewer #1

Abstract

Comment 1: Why is it important to study these bacteria in IEI patients? I suggest that a brief mention of clinical relevance would strengthen the rationale.

Response to comment 1: We sincerely thank you for your insightful and valuable comment, which has helped us strengthen the rationale of our study. We have revised the Abstract, specifically in the background section, to briefly highlight the clinical relevance of studying these bacteria in IEI patients. Specifically, we have added the following content to the Abstract (page 2, lines 5–6, clean version).

“Bacterial infections, especially multidrug-resistant gram-negative bacterial infections pose a great threat to patients with inborn errors of immunity (IEI).”

Comment 2: The authors should give specific WGS techniques or platforms used.

Response to comment 2: We sincerely thank you for your valuable suggestion, which has helped us enhance the methodological clarity of our study. We have added the details of the WGS platform used in our study to the Abstract, specifically in the methods section. The following content was added to the Abstract (page 2, lines 9–11, clean version):

“Strains from IEI and non-IEI groups underwent antimicrobial susceptibility testing and whole-genome sequencing (NovaSeq6000 PE150), with statistical analysis of differences.”

Comment 3: The result section would benefit from mentioning any statistical analyses used to support the comparisons. Were the observed differences statistically significant?

Response to comment 3: Thank you for your insightful suggestion. We have revised the Abstract to briefly mention the statistical analyses conducted to support our comparisons. Fisher's exact test was used to evaluate the differences between the groups. However, due to the small sample size, which is a limitation of our study, most of the observed differences did not reach statistical significance. It is important to note that IEI is a rare condition, and our study focused on drug-resistant strains,

further limiting the sample size. Despite the lack of statistically significant differences, we observed certain trends that may be important for understanding the molecular characteristics of drug-resistant strains isolated from IEI patients. These observations, although descriptive, provide valuable insights into the unique features of these isolates. Specifically, we have added the following content to the Abstract (page 2, lines 9–11 and 21-22, clean version):

“Strains from IEI and non-IEI groups underwent antimicrobial susceptibility testing and whole-genome sequencing (NovaSeq6000 PE150), with statistical analysis of differences.”

“No statistically significant differences were observed between the groups, except for the *fimA* gene in CR-ECO ($p < 0.05$).”

Introduction

Comment 1: I suggest that the authors introduce IEI earlier and clarify how pediatric patients with IEI fit into the broader context of AMR.

Response to comment 1: Thank you for your valuable suggestion. We have revised the Introduction by introducing IEI earlier and clarifying how pediatric patients with IEI fit into the broader context of antimicrobial resistance (AMR). Specifically, we have moved the introduction of IEI to the first paragraph of the Introduction and relocated the discussion of previous reports highlighting the higher AMR observed in immunodeficient patients to the second paragraph, following the introduction of AMR (page 5, lines 2–9 and 20-24, clean version). This revised structure establishes a logical flow, emphasizing the susceptibility of IEI patients to bacterial infections and the high AMR characteristics of the isolates.

Comment 2: The hypothesis can be made more specific than just general bacteria. I suggest a more specific one like 'we hypothesize that carbapenem-resistant bacteria isolated from IEI patients possess distinct genetic characteristics and virulence profiles shaped by their impaired immune environment'.

Response to comment 2: Thank you for your helpful suggestion. We agree that a more specific hypothesis would strengthen the clarity and focus of our study. Accordingly, we have revised the hypothesis to (page 6, lines 10–12, clean version): “We hypothesize that carbapenem-resistant bacteria isolated from patients with IEI possess distinct genetic characteristics and virulence profiles shaped by their impaired immune environment.”

Methods

Comment 1: The authors acknowledge heterogeneity among IEI patients, but they do not mention any stratification or adjustment methods to account for this variability in statistical analyses.

Response to comment 1: Thank you for highlighting this point. We acknowledge the heterogeneity among IEI patients, but due to the small sample size, stratification or adjustment for this variability was not feasible. However, we also compared the major subtypes of IEI patients, including CID and CGD, with the non-IEI group (page 7, lines 17–22, clean version), and unfortunately, the differences were not significant due to the small sample size. Therefore, we mainly use descriptive comparisons and clarify this limitation in the Results and Discussion sections.

Comment 2: The antimicrobial susceptibility testing section lacks details on internal quality controls. Were standard quality control strains (e.g., *E. coli* ATCC 25922, *P. aeruginosa* ATCC 27853) included to verify susceptibility testing accuracy?

Response to comment 2: Thank you for your suggestion. We confirm that standard quality control strains, including *Staphylococcus aureus* ATCC 25923, *Enterococcus faecalis* ATCC 29212, *Escherichia coli* ATCC 25922, and *Pseudomonas aeruginosa* ATCC 27853. This information has been added to the Methods section (page 8, lines 4–7, clean version).

Comment 3: I suggest that the authors should report on assembly quality metrics (e.g., N50 values, genome coverage, contig counts) since these would provide insight into data completeness and accuracy.

Response to comment 3: Thank you for this valuable suggestion. We have now included assembly quality metrics such as clean reads, mapped (%), complete BUSCOs (%), contigs, GC (%), N50, and total length in the Methods section to provide a comprehensive assessment of data completeness and accuracy. This section is added in supplemental **Table S1**.

Comment 4: The robustness of phylogenetic trees is unclear-were bootstrap analyses performed to support tree topology?

Response to comment 4: We appreciate your observation. Bootstrap analyses were indeed performed to assess the robustness of the phylogenetic tree topology. We used 1,000 bootstrap replicates, and this detail has now been added to the Methods section (page 9, lines 12–13, clean version). Additionally, we have updated **Figure 2** to include bootstrap values, ensuring clearer visualization of branch support.

Comment 5: Was a power calculation performed to determine the sample size needed to detect significant differences between IEI and non-IEI groups?

Response to comment 5: Thank you for raising this point. Yes, a power calculation was performed to determine the minimum sample size required to detect significant differences between the IEI and non-IEI groups. We used G*Power 3.1 to conduct an a priori power analysis for a Chi-square test. The results indicated that a **total sample size of 88** was required to achieve sufficient statistical power. However, due to the rarity of IEI and the limited availability of carbapenem-resistant isolates, our final sample size was smaller than the calculated requirement. Nevertheless, the observed effect sizes suggest meaningful differences, and this limitation has been acknowledged in the Discussion section, underscoring the need for further investigation in larger cohorts.

Results

Comment: Some p-values are presented vaguely (e.g., $p < 0.01$), while others are precise ($p = 0.013$). Use a consistent format, preferably providing exact p-values when possible.

Response: Thank you for your helpful suggestion. We have reviewed the p-value

formatting throughout the manuscript. Some p-values are very small, so we present them as $p < 0.001$. The formatting has been corrected to maintain uniformity across all reported results.

Responds to the Reviewer #3

Comment 1: The biological significance of the reduced virulence gene carriage in IEI strains remains speculative—could this be an adaptation to immunocompromised hosts or a trade-off with antimicrobial resistance?

Response to comment 1: Thank you for your insightful observation. We agree that the reduced virulence gene carriage in IEI strains may represent an adaptation to the immunocompromised host environment or a trade-off with antimicrobial resistance.

Theoretically, most IEI patients are likely to have higher antibiotic exposure than immunocompetent children. However, since our study focuses on drug-resistant isolates from IEI patients, both antibiotic pressure and immune pressure may contribute to the altered virulence gene carriage. This study provides a preliminary exploration, and future research should aim to validate these findings using animal models. Nevertheless, the bacteria isolated from IEI patients exhibit distinct molecular characteristics that warrant clinical attention and the implementation of appropriate measures. We have now included a brief discussion of these possible mechanisms in the Discussion section (page 16, lines 16–19, clean version).

Comment 2: The term "immune modulation genes" in CR-AB is somewhat vague. Clarifying the specific pathways or functions these genes are involved in would enhance understanding.

Response to comment 2: Thank you for your suggestion. We have clarified the term "immune modulation genes" by specifying the functions involved. Specifically, these immune regulatory genes are associated with the capsule and play a role in antiphagocytosis. This clarification has been added to the Discussion sections (page 16, lines 7–9, clean version).

Comment 3: The authors acknowledge the heterogeneity of IEI patients included in this study due to the rarity of these diseases. However, it is unclear why CR-AB isolates were specifically analyzed from CID patients, while CR-ECO and CR-ECL were examined in CGD patients. Was this selection based on the prevalence of these bacterial species in each patient group, or were there specific clinical or microbiological reasons for this categorization?

Response to comment 3: Thank you for raising this point. The selection of CR-AB isolates from CID patients and CR-ECO/CR-ECL isolates from CGD patients was based on the prevalence of these bacterial species in each patient group at our center. This distribution also aligns with the types of bacteria to which IEI patients are most susceptible. CR-AB was primarily detected in patients with CID, which is associated with increased susceptibility to severe infections due to impaired T-cell immunity in these individuals. Conversely, CR-ECO and CR-ECL were predominantly detected in patients with CGD, a condition characterized by heightened vulnerability to catalase-positive organisms. We have now clarified this rationale in the Methods section (page 7, lines 17–22, clean version).

Comment 4: It is worth noting that the length of hospital stay for IEI patients was significantly longer than that of non-IEI patients. However, considering the complex underlying conditions in the IEI group, bacterial infections may not have been the primary reason for the extended hospital stay. I would suggest that the authors briefly explain what the possible reasons might be for the longer hospital stay in the IEI group. This would help clarify whether the extended duration was primarily due to the severity of the underlying immune deficiency or other factors.

Response to comment 4: Thank you for your thoughtful suggestion. We acknowledge that the prolonged hospital stay in the IEI group may be attributed not only to bacterial infections but also to their complex disease. Patients with IEI often experience specific infections, including viral, fungal, and mycobacterial infections. Additionally, some patients may present with other severe conditions, such as severe pneumonia, sepsis, acute respiratory distress syndrome, and respiratory failure. As a result, bacterial infections may not have been the primary cause of prolonged hospitalization and were not further explored in this study. We have now included this

explanation in the Discussion section to provide a more comprehensive interpretation of the extended hospitalization duration in the IEI group (page 14, lines 12–17, clean version).

Comment 5: The way percentage ranges are presented in the manuscript (e.g., ((18.2~45.5)% vs. (40.0~80.0)%)) is unconventional and could lead to confusion. To improve clarity, I recommend using a more standard notation, such as (18.2-45.5% vs. 40.0-80.0%).

Response to comment 5: Thank you for pointing this out. We have revised the manuscript to present percentage ranges in a more standard format.

Comment 6: I noticed an inconsistency in the formatting of *p*-values between the main text and Table 1. In the text, the *p*-values are presented in lowercase and non-italicized, while in Table 1, they are capitalized and italicized. Please unify the format.

Response to comment 6: Thank you for highlighting this inconsistency. We have now standardized the formatting of *p* values throughout the manuscript, ensuring that they are consistently presented in lowercase and non-italicized, both in the main text and in Table 1.

Additionally, we have added the running title and Importance section as required by the journal. Beyond the changes requested by the reviewers, we have made the following modifications for clarity and accuracy:

1. In the Methods section, we have moved Table S1 and Table S2 to the Results section (now Table S2 and Table S3). We believe that placing the demographic information in the Results section aligns better with typical reading conventions.

2. In Results section, the original section titled "Antibiotic resistance genes and virulence genes analysis" contained an extensive amount of information. To improve clarity and logical flow, we have restructured it into three separate sections: "Antibiotic resistance and virulence gene profiles of CR-AB strains," "Antibiotic resistance and virulence gene profiles of CR-ECO strains," and "Antibiotic resistance

and virulence gene profiles of CR-ECL strains." This restructuring enhances the clarity and readability of the results section.

3. In the Results section, to better express the content, we have revised the section titles as follows: the original title "Demographic characteristics" has been changed to "Demographic characteristics of patients harboring CR-AB, CR-ECO, and CR-ECL"; the original "Antimicrobial resistance analysis" has been modified to "Antimicrobial resistance of CR-AB, CR-ECO, and CR-ECL isolates"; and the original "MLST typing and phylogenetic analysis" has been revised to "MLST typing and phylogenetic analysis of CR-AB, CR-ECO, and CR-ECL strains".

Re: Spectrum00281-25R1 (Distinct Molecular Characteristics and Virulence Profiles of Carbapenem-Resistant *Acinetobacter baumannii*, *Escherichia coli*, and *Enterobacter cloacae* isolated from Patients with Inborn Errors of Immunity)

Dear Dr. Qinhua Zhou:

Your manuscript has been accepted, and I am forwarding it to the ASM production staff for publication. Your paper will first be checked to make sure all elements meet the technical requirements. ASM staff will contact you if anything needs to be revised before copyediting and production can begin. Otherwise, you will be notified when your proofs are ready to be viewed.

Sincerely,
Bobby Warren
Editor
Microbiology Spectrum

Reviewer #1 (Comments for the Author):

The authors have satisfactorily addressed all my major concerns raised in the initial round of review. They have provided additional clarifications, improved the presentation of their results, and appropriately acknowledged the study's limitations.

Reviewer #3 (Comments for the Author):

the author addressed all of my comments. I agree the revisions.